# A Cross-Sectional Study on the Relationship between Oral Function and Sarcopenia in Japanese Patients with Regular Dental Maintenance

**DOI:** 10.3390/ijerph19095178

**Published:** 2022-04-24

**Authors:** Ryuichi Shirahase, Yutaka Watanabe, Tohru Saito, Yusuke Sunakawa, Yuya Matsushita, Hideki Tsugayasu, Yutaka Yamazaki

**Affiliations:** 1Gerodontology, Department of Oral Health Science, Faculty of Dental Medicine, Hokkaido University, Sapporo 060-8586, Japan; white.wave.ryu@gmail.com (R.S.); sunakawayuusuke@gmail.com (Y.S.); hpnijjq@yahoo.co.jp (Y.M.); yutaka8@den.hokudai.ac.jp (Y.Y.); 2Medical Corporation Shuwa-Kai Tsugayasu Dental Clinic, Obihiro 080-0020, Japan; tohru_saito820@yahoo.co.jp (T.S.); sougou@tsugayasu-sika.jp (H.T.)

**Keywords:** oral hypofunction, sarcopenia, super-aged society, organic dental problems, mobility-impaired dental problems

## Abstract

We aimed to clarify the relationship between oral function assessments regarding oral hypofunction and sarcopenia in patients who had completed treatment for organic dental problems, including oral pain, removable denture fit, and tooth loss. This cross-sectional study included 269 patients aged ≥65 years (mean age 74.9 ± 6.50 years, 133 men, 136 women) who visited a dental clinic between June 2019 and March 2021. We evaluated oral function and sarcopenia and analyzed their relationship using the Jonckheere–Terpstra test, Mantel–Haenszel trend test, and Poisson regression analysis. We diagnosed 132 (49.07%) patients with oral hypofunction, 30 (11.2%) with sarcopenia, and 24 (8.9%) with severe sarcopenia. The number of oral hypofunction items (prevalence rate ratio [PRR] = 1.39, 95%Wald = 0.11 to 0.56) was significantly associated with sarcopenia. For each of the items, tongue-lip motor function [ta] (PRR = 0.80, 95%Wald = −0.44 to −0.02)] [ka] (PRR = 0.76, 95%Wald = −0.53 to −0.03) and tongue pressure (PRR = 0.95, 95%Wald = −0.09 to −0.02) showed a significant association with sarcopenia. However, no significant association was found for other variables. Dentists should not only treat organic dental problems but also consider the relationship between oral function and sarcopenia.

## 1. Introduction

Oral health is well known to be related to general health and quality of life [1]. In recent years, attention has been focused on the decline in oral functions associated with aging [2]. A 2011 cohort study based on community-dwelling older people in Japan reported that poor oral function is a predictor of physical frailty, sarcopenia, disability, and death [3]. In a super-aged society, a society wherein the percentage of the population aged ≥65 exceeds 21% of the total population [4], dentistry in Japan needs to shift from “treatment-oriented” medicine, which mainly restores the form of teeth, to “treatment, management, and coordination-oriented” medicine, that aims to maintain and restore oral functions based on the patient’s life stage [4]. Oral hypofunction is defined as the development of the following seven signs and symptoms: poor oral hygiene, oral dryness, reduced occlusal force, decreased tongue-lip motor function, decreased tongue pressure, decreased masticatory function, and decreased swallowing function; additionally, oral hypofunction is diagnosed when three or more of these symptoms are present [5]. It has been included in the Japanese medical insurance system as a new disease name to promote this shift.

Previous studies have shown that people with reduced oral function tend to avoid foods that are difficult to chew and are, thus, more susceptible to malnutrition [6]. It has also been reported that as oral function declines, opportunities for exercise decrease [7]. Poor nutrition and inadequate exercise are both causes of sarcopenia, and early detection and intervention are important for improving prognosis [8]. Sarcopenia is defined as “age-related loss of skeletal muscle mass, muscle strength, and/or physical performance” [9]. It is a risk factor for disability, hospitalization, death, and dementia [9]. Some previous studies have investigated the relationship between oral function and sarcopenia [10,11,12,13]. However, most of them are single oral function assessments, and only a few have investigated the relationship between poor oral function and sarcopenia using all the items of the oral hypofunction test [14,15]. Oral pain, prosthetic status for missing teeth, or the fit of removable dentures may also affect oral function assessment. However, previous studies have not considered dental diseases, such as dental caries, missing teeth, or poor fit of removable dentures in participants [3,10,11,12,13,14,15]. Kikutani defined organic masticatory disorder as a masticatory disorder caused by loss of masticatory organs due to pain in the mouth, progression of periodontal disease, and others, and mobility-impaired impaired masticatory disorder as a masticatory disorder caused by loss of oral function due to aging or disease [16]. According to him, oral problems can be broadly divided into organic problems and mobility-impaired problems. We believe that investigating the association between oral hypofunction and sarcopenia after excluding the effects of organic dental problems will further clarify that functional decline due to mobility-impaired dental problems is associated with sarcopenia. It will strengthen the evidence that declining oral function is related to general health. It also demonstrates the possibility that dental treatment aimed at maintaining and restoring oral function can contribute to general health. We hypothesized that oral hypofunction is associated with sarcopenia even in participants who are treated with restorative or prosthetic treatment for dental problems and are less affected by organic dental problems. Therefore, this study aimed to clarify the relationship between oral function assessment for oral hypofunction and sarcopenia in patients who had completed treatment for organic dental problems.

## 2. Materials and Methods

### 2.1. Design and Participants

This cross-sectional study included outpatients aged ≥65 years who visited community dental clinics in a major city in northern Japan from June 2019 to March 2021. The study participants were recruited from patients who visited the clinic for regular dental maintenance every 1–3 months or who transitioned to regular maintenance after completing treatment. Treatment completion was defined as the absence of untreated caries, completion of initial periodontal treatment, occlusal support on both molars by natural or prosthetic teeth, and no complaints of difficulty in chewing. If patients complained of dental problems during regular dental maintenance, problem resolution was a priority. Patients were included in the study after satisfying the defining criterion of treatment completion. Those with impaired physical function due to comorbidities or injuries, pacemakers, and a history of head and neck cancer were excluded from the study. The contents of the study were explained to the patients verbally and in writing before written consent was obtained. The age, sex, height, weight, and comorbidities of the patients were assessed and recorded. The study included a total of 294 patients who consented to participate. However, 25 patients were excluded: 12 had difficulty walking independently, one had a pacemaker, two had difficulty measuring grip strength due to hand injury, and ten had missing data. The remaining 269 patients (mean age 74.9 ± 6.50 years, 133 men and 136 women) were considered study participants. The adjustment variables were selected in advance using a directed acyclic graph (Figure 1).

This study was conducted in accordance with the Declaration of Helsinki and was approved by the Ethical Review Committee for Clinical and Epidemiological Research, Hokkaido University School of Dentistry (Approval No. 2019-4).

### 2.2. Survey Items

Oral examination (number of functioning teeth, use of removable dentures, and oral function assessment), gait speed, grip strength, and body composition were measured.

### 2.3. Oral Assessment

Surveys on the number of functional teeth and the use of removable dentures and oral function assessments were conducted by seven dental hygienists from clinics that had undergone prior training and standardization. The number of functional teeth was defined as the sum of the number of remaining teeth and the number of pontics in the bridge, the number of dental implants, and the number of artificial teeth in removable dentures. Teeth with only the root remaining and tooth mobility degree III were excluded from the number of functional teeth [17]. It was also confirmed that the patient had not eaten, drunk, brushed, or gargled within one hour prior to the assessment.

#### 2.3.1. Oral Hygiene

Oral hygiene was evaluated by visual examination using the tongue coating index (TCI) [18], which is an index of the degree of tongue coating. This index divides the surface of the tongue into nine areas, evaluates the degree of tongue coating in each area based on a three-point scale (score 0, 1, 2), and calculates the total score. Participants with a TCI of 50% or more (a total score of 9 or more) were defined as having poor oral hygiene.

#### 2.3.2. Oral Dryness

Oral dryness was evaluated based on the degree of oral mucosal wetness at the center of the tongue, approximately 10 mm from the tongue tip, using an oral moisture meter (Mucus, Life, Saitama, Japan) [19]. Measurements were obtained three times, and the median value was used as the measurement value. A value of less than 27.0 indicated oral dryness.

#### 2.3.3. Occlusal Force

The occlusal force was evaluated according to the number of remaining teeth. Teeth with only the roots remaining, tooth mobility degree III, bridge pontics, and dental implants were excluded from the number of remaining teeth. A decreased occlusal force was defined as less than 20 remaining teeth [5].

#### 2.3.4. Tongue-Lip Motor Function

Participants were required to repeat the syllables [pa], [ta], and [ka] as quickly as possible within 5 s and the number of times each syllable was pronounced in 1 s was measured using an automatic measuring device (Kenko-kun Handy, Takei Kiki Kogyo, Niigata, Japan) [20]. Participants who pronounced any of the syllables less than six times were defined as having decreased tongue-lip motor function.

#### 2.3.5. Tongue Pressure

Tongue pressure was evaluated using an appropriate device (JMS Tongue Pressure Measuring Device, JMS, Hiroshima, Japan) [21]. Participants were instructed to position the hard ring part of the tongue pressure probe between the maxillary central incisors and push the pressure-receiving part (balloon) up to the palate using the tongue at the maximum force with their lips closed. For those with removable dentures, measurements were taken with their dentures inserted. Measurements were obtained three times, and the average value was used as the measurement value. A value of less than 30 kPa indicated low tongue pressure.

#### 2.3.6. Masticatory Function

Participants were required to freely chew 2 g of gummy jelly (Glucolum, G.C., Tokyo, Japan) for 20 s. They were subsequently required to rinse their mouths with 10 mL of water, and the gummies and water were spat out into a filtration mesh. The amount of glucose eluted in the solution that passed through the mesh was measured using the masticatory ability test system (Glucosensor GS-II, G.C., Tokyo, Japan) [22] to determine the concentration of eluted glucose. A glucose concentration of less than 100 mg/dL was defined as a decrease in masticatory ability.

#### 2.3.7. Swallowing Function

Swallowing function was assessed using the Seirei dysphagia screening questionnaire [23], which included 15 items. A minimum of one “A” response was defined as having decreased swallowing function.

All the tests (Section 2.3.1, Section 2.3.2, Section 2.3.3, Section 2.3.4, Section 2.3.5, Section 2.3.6 and Section 2.3.7) were performed, and patients who had decreased oral function in three or more tests were diagnosed with oral hypofunction.

### 2.4. Sarcopenia Assessment

Sarcopenia was assessed by three trained registered dietitians from the clinic, and all tests were standardized in advance.

#### 2.4.1. Muscle Strength

A Smedley grip strength meter (electronic hand dynamometer, SODIAL(R), Shenzhen, China) was used to measure the grip strength of the patients’ dominant hand. The measurements were taken twice, and the maximum value was recorded. Low muscle strength was defined as less than 28 kg for men and less than 18 kg for women.

#### 2.4.2. Physical Performance

The time required to pass a 6-m straight walking path was measured while participants were walking at a normal pace. A 1-m running section was provided before and after the measurement area. Gait speed was calculated from the measured values, and a speed of less than 1 m/s was defined as a decline in physical function.

#### 2.4.3. Appendicular Skeletal Muscle Mass

Appendicular skeletal muscle mass was measured using a body composition analyzer, InBody470 (InBody Japan, Tokyo, Japan). Skeletal muscle mass loss was defined as less than 7.0 kg/m^2^ in men and less than 5.7 kg/m^2^ in women. Based on the results from Section 2.4.1, Section 2.4.2 and Section 2.4.3, the participants were classified into three groups, including the normal, sarcopenia, and severe sarcopenia groups, according to the Asian Working Group for Sarcopenia 2019 (AWGS2019) criteria [8] (Figure 2).

### 2.5. Statistical Analysis

The appropriate sample size was calculated using G*Power 3.1.9.7 software (Kiel University, Kiel, Germany). The effect size was set to medium, with an α error of 0.05, power of 0.80, and N2/N1 of 0.164 based on a previous study [14] that reported the prevalence of sarcopenia to be 16.4%. The required number of participants was 206. IBM SPSS Statistics 27 (IBM, New York, NY, USA) was used for statistical analyses, and the statistical significance level was set at *p* < 0.05 and 95%Wald, not including 1. The Kolmogorov–Smirnov normality test was performed. Kendall’s Tau-b test was conducted to evaluate the correlation between each test item in the oral function assessment. The Jonckheere–Terpstra and Mantel–Haenszel trend tests were used to investigate trends among participants in the normal, sarcopenia, and severe sarcopenia groups. The association of general health conditions (age, sex, body mass index [BMI], hypertension, diabetes mellitus, hyperlipidemia, cerebrovascular disease, malignancy, and cardiovascular disease) [3,24,25,26] with sarcopenia was investigated using Poisson regression analysis. To investigate the relationship between sarcopenia and oral hypofunction, two Poisson regression models were constructed: Model 1 was a series of univariate models with the presence or absence of sarcopenia (two groups: normal sarcopenia and severe sarcopenia) as the dependent variable, while the nine oral function assessments, diagnosis of oral hypofunction, number of oral hypofunction items, and removable denture use were used as the independent variables. In contrast, Model 2 was a multivariate model adjusted with the significant variables found in the univariate models.

## 3. Results

According to the results presented in Table 1, which shows the number and percentage of patients for each test item, the minimum number of participants for each test item was 21 (7.8%) for masticatory function, and the maximum number was 170 (63.2%) for tongue-lip motor function. Of the 269 participants, 132 (49.07%) were diagnosed with oral hypofunction, 30 (11.2%) with sarcopenia, and 24 (8.9%) with severe sarcopenia.

Table 2 shows trends in participant characteristics among the three groups. The highest incidence of oral hypofunction diagnosis and number of oral hypofunction items were observed in the severe sarcopenia group, followed by the sarcopenia group and then the normal group. Oral function tended to decrease with increasing sarcopenia severity compared to that in the normal group. Significant differences were observed in the number of remaining teeth, tongue-lip motor function [pa][ta][ka], tongue pressure, masticatory function, and swallowing function.

Table 3 shows the results of the Poisson regression analysis of the association between sarcopenia and age, sex, and medical history. Significant associations were observed between age and BMI.

Table 4 shows the relationship between sarcopenia and each oral function. In univariate analyses, there was a significant association between sarcopenia and the number of remaining teeth, tongue-lip motor function [pa][ta][ka], tongue pressure, masticatory function, and the number of oral hypofunction items. In the second column, Model 2, age, gender, presence of diabetes mellitus, and the presence of cerebrovascular disease were used as covariates. After selection variables were applied with the covariates, tongue-lip motor function [pa] (prevalence rate ratio [PRR] = 0.80, 95%Wald = −0.44 to −0.02) [ka] (PRR = 0.76, 95%Wald = −0.53 to −0.03), tongue pressure (PRR = 0.95, 95%Wald = −0.09 to −0.02), and number of oral hypofunction items (PRR = 1.39, 95%Wald = 0.11 to 0.56) were significantly associated with sarcopenia.

## 4. Discussion

This study showed that the prevalence of sarcopenia was significantly higher as the number of oral hypofunction items increased in outpatients who had completed treatment for organic dental problems. This indicates an association between sarcopenia and poor oral function even in patients who receive appropriate dental treatment for organic dental problems and do not have oral pain or occlusal support deficits.

Previous studies suggested an association between oral function and sarcopenia [14,15]. Kugimiya et al. conducted a study that investigated all items of oral function assessments on older people living in a community, and a significant association was observed between oral hypofunction and sarcopenia [15].

Our results support those of previous studies; however, Nakamura et al. found no significant association between oral hypofunction and sarcopenia in a study of older people living in a community [14]. In that study, subjective evaluation of masticatory ability was conducted using a questionnaire; however, it was reported that subjective and objective evaluations of masticatory ability do not always coincide [27]. In addition, these previous studies were conducted on older people living in the community and did not consider the status of dental problems, such as dental caries, missing teeth, and ill-fitting removable dentures, which may have been affected by oral pain and loss of occlusal support.

To our knowledge, this is the first study showing the relationship between oral function and sarcopenia by conducting a full assessment of oral hypofunction in patients who had completed treatment for dental diseases and were less affected by organic dental problems. Ikebe et al. reported that oral hypofunction is present in 40–50% of older community residents [28]; the present study results were similar (49.1%). Regarding sarcopenia, a study conducted using the AWGS2019 in older people living in the community reported that 14.4% had sarcopenia, and 4.2% had severe sarcopenia [15].

Although this study had a biased population as it included patients from a single dental clinic, the prevalence of oral hypofunction and sarcopenia was comparable to that of previous studies.

In addition, the oral problems of the present study participants were treated following a consistent policy. Conservative treatment was performed for defective tooth structure and prosthodontic treatment for tooth loss. Oral hygiene was also managed by dental hygienists who had received the same in-office training. The number of functional teeth was 27.53 ± 1.01. When compared to a previous study of older people living in the community [29], the present study showed that the number of functional teeth was close to the mean but with a smaller standard deviation. In this study, only 7.8% of patients had decreased masticatory function, a proportion which is small compared to those observed in a previous study [28].

Therefore, we believe that the present study participants were less affected by organic dental problems than those in previous studies [14,15], and this is a characteristic of this study.

Focusing on each item of oral function assessment, the tongue-lip motor function and tongue pressure were significantly related to sarcopenia. The relationship between tongue-lip motor function and sarcopenia, and between tongue pressure and sarcopenia have already been reported in a previous study [11], consistent with our results.

In this study, oral hygiene, oral dryness, number of remaining teeth, masticatory function, and swallowing function were not significantly associated with sarcopenia. Specifically, oral hygiene was not significantly associated with sarcopenia because the participants in this study received oral hygiene instructions from a dental hygienist as part of their dental care. We did not find any reports of an association between oral dryness and sarcopenia. The number of remaining teeth was found to be significantly associated with sarcopenia in a previous study of older people living in the community [10,13,30]. Since the participants in this study included those with restored occlusal support, we considered that the association between the number of remaining teeth and sarcopenia was weakened.

The relationship between sarcopenia and masticatory function had already been reported to be significant in previous studies [10]. In this study, only 7.8% of the patients had decreased masticatory function, which is less than that observed in previous studies [10,28] and may be attributed to the fact that this study was conducted on a population with restored occlusal support. Therefore, the small number of patients with decreased masticatory function may be the reason for the inconsistency in the results. Swallowing function was significantly associated with sarcopenia in a previous study of community-dwelling older women [31] and a previous study of hospitalized patients [32]. The study of community-dwelling older women [31] comprised a population with a high mean age of 82.3 ± 6.9 years. The study of hospital inpatients [32] showed a higher prevalence of sarcopenia than that observed in this study. These differences in age and sarcopenia prevalence may have influenced the association between swallowing function and sarcopenia.

Appropriate dental treatment for dental caries, periodontal disease, and tooth loss to resolve organic oral problems has been reported to be effective in preventing malnutrition, a risk factor for sarcopenia [33]. However, the prevalence of sarcopenia was higher in those with reduced oral function, despite having completed treatment for oral problems.

This result indicates that restoration of tooth shape alone is not sufficient to maintain the health of older people against oral problems. The results suggest that treatment aimed at maintaining and improving oral functions, such as oral function training [34] and nutritional guidance [35], is also necessary for mobility-impaired dental problems to prevent sarcopenia. In contrast, the presence of sarcopenia may have affected the decline in oral function. If oral function decline persists after treatment for organic dental problems, the presence of sarcopenia should be suspected, and skeletal muscle mass, muscle strength, and physical function assessments should be considered, as well as dietary and exercise therapy [36].

This study had some limitations. First, since this was a cross-sectional study, we could not determine the causal relationship between oral hypofunction and sarcopenia. To verify their causal relationship, it is necessary to conduct an intervention study wherein patients with oral function decline are classified into oral function training and non-training groups. However, we conducted a cross-sectional study because we thought it was necessary to first understand the actual situation. Second, the number of participants in this study was limited because it only included patients who completed treatment at a single dental clinic in a specific area; hence, bias exists because the response to organic dental disorders is based on the treatment guidelines of a single dental institution. Therefore, the sample size of this study was small compared to that of previous studies [14,15], and the independent variables were limited. Third, we did not investigate lifestyle habits, such as alcohol consumption, smoking, or exercise habits, which are reportedly associated with sarcopenia or the presence or absence of a cohabitant [37,38,39,40]. The possibility that these factors may have affected the results should not be ruled out. Fourth, prosthetic analysis was not performed in this study because it was difficult to categorize the results due to the variation in tooth loss sites and ranges. Fifth, this study included participants with fewer than 20 remaining teeth. To completely exclude the effects of organic dental problems, only those with more than 20 teeth should be included in the study. However, approximately half of the older population in Japan has fewer than 20 teeth [28]. If participants with less than 20 remaining teeth were excluded, the results would deviate greatly from the actual situation; therefore, those with less than 20 remaining teeth were included in the study. Finally, although the study participants had completed initial periodontal treatment, we did not investigate their detailed periodontal status. Although there is no unified view on the relationship between periodontal disease and sarcopenia, previous studies have reported on the relationship between systemic inflammation and sarcopenia [41], which may have influenced the results.

We believe that additional research on these factors using a larger sample size is necessary to strengthen our findings.

## 5. Conclusions

There is a significant association between poor oral function and sarcopenia, even in older people whose treatment of organic dental problems has been completed and have no ongoing dental complaints such as pain. Japan is in the midst of a super-aged society, wherein the structure of dental diseases is also changing along with the alterations in population structure. Dental treatment in a super-aged society should not be limited to organic treatment but should also consider the relationship between mobility-impaired dental problems and sarcopenia.

## Figures and Tables

**Figure 1 ijerph-19-05178-f001:**
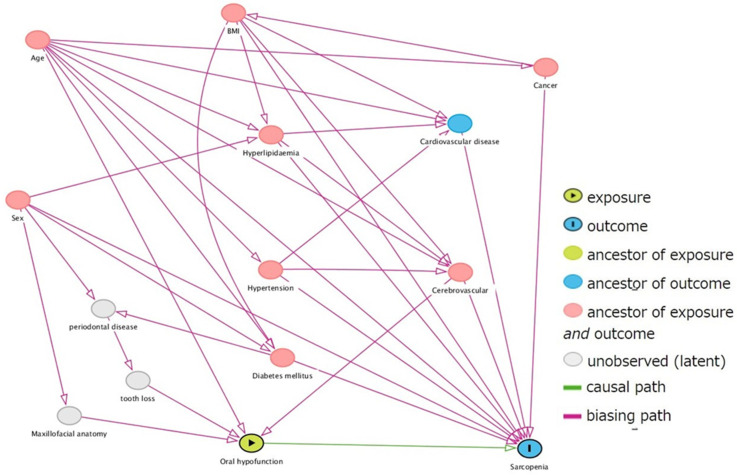
Selection of adjustment variables by directed acyclic graph.

**Figure 2 ijerph-19-05178-f002:**
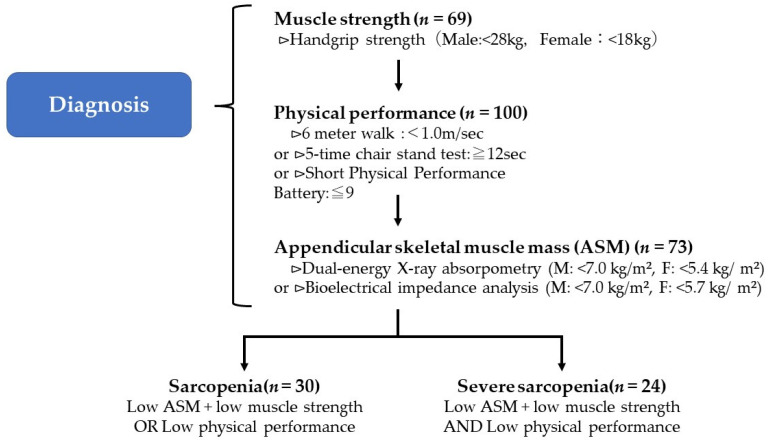
AWGS2019 criteria for the diagnosis of sarcopenia.

**Table 1 ijerph-19-05178-t001:** Items for oral function assessment and the number and percentage of patients with oral hypofunction.

Clinical Signs	Measurement	Number	%
Poor oral hygiene	Tongue coating index ≥ 50%	52	19.33
Oral dryness	The measured value obtained by a recommended moisture checker is less than 27.0.	115	42.75
Reduced occlusal force	The number of remaining teeth is less than 20.	131	48.70
Decreased tongue-lip motor function	The number of /pa/, /ta/ or /ka/ pronunciations per second is less than 6.	170	63.20
Decreased tongue pressure	The maximum tongue pressure is less than 30 kPa.	130	48.33
Decreased masticatory function	The glucose concentration obtained by chewing gelatin gummies is less than 100 mg/dL.	21	7.80
Decreased swallowing function	The A items on the Seirei dysphagia screening questionnaire more than 1.	68	25.30

**Table 2 ijerph-19-05178-t002:** Trends in participant characteristics among the study groups.

	All	Normal	Sarcopenia	Severe Sarcopenia	*p*-Value	Analysis
Variables	(*n* = 269)	(*n* = 215)	(*n* = 30)	(*n* = 24)		
**Characteristics**						
Age	74.93 ± 6.50	73.94 ± 6.03	77.17 ± 6.64	80.96 ± 6.62	<0.001 ***	a
Sex (%, male)	49.44	46.05	56.67	70.83	0.016 *	b
Height (cm)	157.22 ± 8.79	158.01 ± 8.77	152.45 ± 7.22	155.41 ± 8.83	0.005 **	a
Weight (kg)	59.05 ± 10.61	60.58 ± 9.76	52.04 ± 12.45	54.10 ± 10.91	<0.001 ***	a
BMI (kg/m^2^)	23.59 ± 3.29	24.14 ± 3.22	21.52 ± 3.03	21.25 ± 1.97	<0.001 ***	a
**Variables of sarcopenia**						
Grip strength (kg)	28.13 ± 11.96	29.44 ± 11.89	25.83 ± 13.01	19.31 ± 5.80	<0.001 ***	a
Gait speed (m/s)	1.06 ± 0.27	1.10 ± 0.26	1.02 ± 0.23	0.78 ± 0.23	<0.001 ***	a
SMI (kg/m^2^)	6.67 ± 0.96	6.85 ± 0.92	5.85 ± 0.80	5.99 ± 0.66	<0.001 ***	a
**Variables of oral hypofunction and oral health status**						
Oral hygiene (%)	32.48 ± 17.26	32.36 ± 16.87	32.15 ± 21.36	34.03 ± 15.66	0.844	a
Oral dryness	26.91 ± 3.09	27.03 ± 3.09	26.87 ± 2.80	25.94 ± 3.43	0.136	a
Number of remaining teeth	17.25 ± 8.81	17.99 ± 8.41	14.13 ± 9.83	14.63 ± 9.96	0.016 *	a
Tongue-lip motor function/pa/ (times/s)	6.09 ± 0.96	6.23 ± 0.83	5.75 ± 0.99	5.36 ± 1.51	<0.001 ***	a
/ta/ (times/s)	6.04 ± 0.93	6.15 ± 0.86	5.68 ± 0.83	5.48 ± 1.34	<0.001 ***	a
/ka/ (times/s)	5.92 ± 0.95	5.75 ± 0.85	5.17 ± 0.95	4.98 ± 1.32	<0.001 ***	a
Tongue pressure (KPa)	30.27 ± 8.27	31.50 ± 7.88	27.65 ± 8.56	22.51 ± 6.40	<0.001 ***	a
Masticatory function (mg/dL)	182.87 ± 59.53	187.37 ± 59.88	161.30 ± 39.30	169.50 ± 70.74	0.013 *	a
Swallowing function (%) ^†^	25.29	21.40	30.00	54.17	0.001 **	b
Number of oral hypofunction items	2.55 ± 1.31	2.34 ± 1.21	3.03 ± 1.19	3.75 ± 1.51	<0.001 ***	a
Diagnosis of oral hypofunction (%)	49.07	43.72	64.52	75.00	0.001 **	b
Number of functional teeth	27.53 ± 1.01	27.56 ± 0.96	27.50 ± 1.22	27.29 ± 1.16	0.698	a
Removable denture use (%)	64.68	64.65	64.52	62.50	0.926	b
**Medical history**						
Hypertension (%)	26.77	27.91	16.13	29.17	0.677	b
Diabetes mellitus (%)	12.27	9.30	16.13	33.33	0.001 **	b
Hyperlipidemia (%)	12.27	12.56	9.68	12.50	0.865	b
Cerebrovascular disease (%)	6.32	4.65	9.68	16,67	0.014 *	b
Cardiovascular disease (%)	11.15	11.63	6.45	12.50	0.269	b
Cancer (%) ^‡^	5.95	6.05	9.68	0.00	0.498	b

BMI, body mass index; SMI, skeletal muscle mass index. a. Jonckheere–Terpstra trend test. b. Mantel–Haenszel trend test. * *p* < 0.05, ** *p* < 0.01, *** *p* < 0.001. ^†^ Percentage of patients with reduced swallowing function. ^‡^ Exclude head and neck cancer.

**Table 3 ijerph-19-05178-t003:** Relationship between sarcopenia and age, BMI, sex, and medical history.

Variables	PPR	95%Wald		*p*-Value
		Lower Limit	Upper Limit	
Age	1.09	0.05	0.12	<0.001 ***
BMI	0.81	−0.30	−0.13	<0.001 ***
Sex	1.73	−0.01	1.10	0.053
Hypertension	0.78	−0.89	0.39	0.443
Diabetes mellitus	2.34	0.23	1.47	0.008 **
Hyperlipidemia	0.89	−0.97	0.73	0.788
Cerebrovascular disease	2.20	−0.01	1.58	0.052
Cancer	0.93	−1.24	1.09	0.898
Cardiovascular disease	0.81	−1.13	0.71	0.653

** *p* < 0.01, *** *p* < 0.001. Poisson regression analysis was performed. Dependent variable: two sarcopenia groups. Independent variables: age, BMI, sex, hypertension, diabetes, dyslipidemia, cerebrovascular disease, cancer, and cardiovascular disease. PRR. prevalence rate ratio.

**Table 4 ijerph-19-05178-t004:** Relationship between sarcopenia and oral function assessment, removable denture use, number of functional teeth, diagnosis of oral hypofunction, and number of oral hypofunction items.

Variables	Model 1				Model 2			
	95% Wald				95% Wald			
	PRR	Lower Limit	Upper Limit	*p*-Value	PRR	Lower Limit	Upper Limit	*p*-Value
**Oral hygiene**	1.00	−0.01	0.02	0.816	1.00	−0.02	0.02	0.861
**Oral dryness**	0.96	−0.12	0.04	0.281	0.97	−0.11	0.05	0.404
**Number of remaining teeth**	0.97	−0.06	−0.07	0.017 *	0.99	−0.05	0.02	0.359
**Tongue-lip motor function /pa/**	0.69	−0.56	−0.19	<0.001 ***	0.80	−0.44	−0.02	0.035 *
** /ta/**	0.68	−0.61	−0.17	<0.001 ***	0.80	−0.48	0.02	0.070
** /ka/**	0.64	−0.65	−0.23	<0.001 ***	0.76	−0.53	−0.03	0.029 *
**Tongue pressure**	0.93	−0.10	−0.04	<0.001 ***	0.95	−0.09	−0.02	0.003 **
**Masticatory function**	1.00	−0.01	−0.00	0.027 *	1.00	−0.01	0.00	0.207
**Swallowing function**	2.02	0.16	1.25	0.011 *	1.60	−0.10	1.04	0.108
**Removable denture use**	1.01	−0.55	0.57	0.968	0.81	−0.79	0.36	0.467
**Number of functional teeth**	0.90	−0.33	0.13	0.374	0.89	−0.34	0.10	0.296
**Diagnosis of oral hypofunction**	2.45	0.31	1.48	0.003 **	1.63	−0.14	1.12	0.127
**Number of oral hypofunction items**	1.57	0.25	0.65	<0.001 ***	1.39	0.11	0.56	0.004 **

Model 1. Poisson logistic regression analysis was performed. Dependent variable: two sarcopenia groups. Independent variables: each item of the oral function assessment and the number of functional teeth, removable denture use, diagnosis of oral hypofunction, and the number of oral hypofunction items. Model 2. In addition to model 1, age, sex, diabetes mellitus, and cerebrovascular disease were entered as covariates. * *p* < 0.05, ** *p* < 0.01, *** *p* < 0.001.

## Data Availability

Not applicable.

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
