# Peer review of "A Cross-Sectional Study on the Relationship between Oral Function and Sarcopenia in Japanese Patients with Regular Dental Maintenance"

_ijerph, 2022, doi:10.3390/ijerph19095178_

Round 1
Reviewer 1 Report
Dear Authors
The manuscript is very interesting.
I would really appreciate authors for their thinking process and creating new way of presentation.
In introduction I would like suggest some more data on need of the study .
Methods, since it appears some confusion need clarity on inclusion and exclusion criteria statistical analysis: kindly make it more clear on comparisons is not very clear in analysis
results: clearly stated.
discussion: need update recent findings and comparison of same oral findings w with similar conditions.
conclusions: Based on objective.
reference: need to update:
overall great news work.
Author Response
Thank you very much for reviewing our article in detail despite the difficult situation caused by COVID-19. We have thoroughly reviewed your kind peer-review comments, all of which we agree with, and have revised the article accordingly.
Comment: In introduction I would like suggest some more data on need of the study .
Response: Thank you for your advice. We have cited additional references in the Introduction to emphasize the need for this study. (references 12 and 13)
Comment: Methods, since it appears some confusion need clarity on inclusion and exclusion criteria
Response: We agree with your advice and have added a description of the study participants to the manuscript. (page 2, lines 78–86).
Comment: statistical analysis: kindly make it more clear on comparisons is not very clear in analysis.
Response: Thank you for your advice. Table 2 shows trends in participant characteristics among the three study groups: normal, sarcopenia, and severe sarcopenia groups. The word "compair" may not sufficiently convey our intended meaning; hence, we have revised it for improved readability. (page 6, lines 194, 212–215, and title of Table 2)
Comment: discussion: need update recent findings and comparison of same oral findings w with similar conditions.
Response: Following your advice, I have added as a reference a study published in 2021 on the association between the number of remaining teeth and sarcopenia. (references 10 and 13)
Comment: reference: need to update:
Response: Thank you for your advice. References have been revised to reflect the above changes.
Please check the manuscript, which is attached.

Reviewer 2 Report
Summary of manuscript: This manuscript aims to the study of the relationship between oral function assessment for oral hypofunction and sarcopenia in patients who had completed treatment for organic dental problems. This report concludes that there is a significant association between poor oral function and sarcopenia, even in the elderly whose treatment of organic dental problems has been completed.
General comments
- The authors should reexamine the study design to clarify that sarcopenia is caused by oral hypofunction.
Overall manuscript
- Please proof-read the manuscript by a native English speaker.
Author Response
This study does not aim to identify sarcopenia is caused by oral hypofunction.
This study aims to clarify the association between sarcopenia and oral hypofunction in a cross-sectional study. (page 2, lines 71-73)
We believe our study design is appropriate.
In addition, we have received native proofreading of the English text and have attached a certificate of such proof.

Reviewer 3 Report
This manuscript is of novel interest to investigate the relationship between oral function and sarcopenia. A total of 269 patients and half are diagnosed with sarcopenia. However, some edits may be needed to strengthen the manuscript.
- "Organic dental problem" is not very clear and should be revised with scientific writing and term
- And what intervention or specific care should be implented for those population with sarcopenia and oral hypofunction?
- What's the sampling and recruiting method for the 260 patients?
Author Response
Thank you very much for reviewing our article in detail despite the difficult situation caused by COVID-19. We have thoroughly reviewed your kind peer-review comments, all of which we agree with, and have revised the article.
Comment: Title: "Organic dental problem" is not very clear and should be revised with scientific writing and term
Response: Thank you for your comments. “Organic dental problems” is an extensive term for “organic masticatory disorder,” as described by kikutani et al. (Reference. 16). In the text, the term is used after defining it. (page 2, lines 59–63) As you indicated, "organic dental problems" may not be an appropriate term to use in the title; hence, the title has been revised as below:
A cross-sectional study on the relationship between oral function and sarcopenia in Japanese patients with regular dental maintenance.
Comment: And what intervention or specific care should be implented for those population with sarcopenia and oral hypofunction?
Response: Thank you for indicating this point. We believe that patients with sarcopenia and oral function benefit from oral function training and nutritional guidance. We also consider that the presence of sarcopenia should be suspected when oral functional decline is observed after dental treatment has been completed. Accordingly, the above information has been mentioned in the text. (page 10, lines 308–316)
Comment: What's the sampling and recruiting method for the 260 patients?
Response: Thank you for indicating this point. Accordingly, we have added a description of the study participants to the manuscript. (page 2, lines 78–86).
Please check the manuscript, which is attached.

Round 2
Reviewer 1 Report
All the quires have been addressed.